# Message-Passing GNNs Fail to Approximate Sparse Triangular Factorizations

**Vladislav Trifonov**  *vladtrifono@gmail.com*
*AIC, Skoltech*
*AI4S Center, Sberbank of Russia*

**Ekaterina Muravleva**  *e.muravleva@skoltech.ru*
*AI4S Center, Sberbank of Russia*
*AIC, Skoltech*

**Ivan Oseledets**  *oseledets@airi.net*
*AIRI*
*AIC, Skoltech*

**Reviewed on OpenReview:** *https://openreview.net/forum?id=YIr9SzD3C9*

## Abstract

Graph Neural Networks (GNNs) have been proposed as a tool for learning sparse matrix preconditioners, which are key components in accelerating linear solvers. We present theoretical and empirical evidence that message-passing GNNs are fundamentally incapable of approximating sparse triangular factorizations for classes of matrices for which high-quality preconditioners exist but require non-local dependencies. To illustrate this, we construct a set of baselines using both synthetic matrices and real-world examples from the SuiteSparse collection. Across a range of GNN architectures, including Graph Attention Networks and Graph Transformers, we observe low cosine similarity ($\leq 0.7$ in key cases) between predicted and reference factors. Our theoretical and empirical results suggest that architectural innovations beyond message-passing are necessary for applying GNNs to scientific computing tasks such as matrix factorization. Moreover, experiments demonstrate that overcoming non-locality alone is insufficient. Tailored architectures are necessary to capture the required dependencies since even a completely non-local Global Graph Transformer fails to match the proposed baselines.

## 1 Introduction

Preconditioning sparse symmetric positive definite matrices is a fundamental problem in numerical linear algebra (Benzi, 2002). The goal is to find a precondtioner matrix $X$ such that $X^{-1}A$ is well-conditioned. This results in faster convergence of iterative methods when solving linear systems (Saad, 2003). A well-established choice for symmetric positive definite matrices is to use an incomplete Cholesky factorization as the direct preconditioner $X = LL^\top \approx A$. Recently, there has been significant interest in using graph neural networks (GNNs) to predict such preconditioners (Li et al., 2024; Trifonov et al., 2024; Häusner et al., 2023; 2024). The key idea is to represent the sparse matrix $A$ as a graph where edges correspond to the non-zero entries and to use GNN architectures to predict the entries of the preconditioner $X$, minimizing a certain functional.

Although GNNs show promise, we demonstrate their fundamental limitation in tasks related to solving linear systems: their locality prevents them from learning non-local preconditioners. Specifically, we demonstrate that there are classes of matrices for which optimal sparse preconditioners exist, yet they exhibit non-local

dependencies. Starting with simple matrices, such as tridiagonal matrices arising from a discretization of partial differential equations (PDEs), we demonstrate that updating a single entry in $A$ can significantly affect all entries in $L$.

To promote further development in the field, we introduce a new benchmark dataset of matrices for which optimal sparse preconditioners are known to exist but require non-local computations. We construct this dataset using both synthetic examples and real-world matrices from the SuiteSparse collection (Davis, 2024). For the synthetic benchmarks, we carefully design tridiagonal matrices for which the Cholesky factors depend non-locally on the matrix elements by leveraging properties of rank-1 semiseparable matrices. For real-world problems, we explicitly compute so-called K-optimal preconditioners based on the inverse matrix with sparsity patterns matching the lower triangular part of the original matrices.

Although message-passing GNNs can naturally be applied to sparse linear algebra problems, their inherent locality poses fundamental limitations in learning non-local linear algebra operations such as preconditioner construction. Our contribution is as follows:

- We demonstrate the fundamental limitations of message-passing GNNs in approximating generally non-local preconditioner matrices for sparse linear systems.

- We construct a benchmark dataset to validate novel architectures for GNN-based preconditioner construction routines and to validate novel non-local GNN architectures. The dataset consists of both synthetic and real-world matrices. The real-world matrix dataset is based on the solution to an optimization problem, resulting in optimal sparse preconditioners.

- We demonstrate that specific architectures are required for the task of constructing GNN-based preconditioners, since even completely non-local architectures, such as graph transformers, fail to approximate the proposed benchmark, indicating the issue is architectural mismatch, not solely insufficient receptive field.

## 2   Problem Formulation

Let $A$ be a sparse symmetric positive definite matrix. The goal is to find a sparse lower triangular matrix $L$ such that $LL^\top$ approximates $A$ well. In other words, the condition number of $L^{-\top}AL^{-1}$ is small. Moreover, a good preconditioner should produce a tightly clustered spectrum of $L^{-\top}AL^{-1}$. The key idea in approximating $L$ with GNNs is representing the sparse matrix $A$ as a graph where the edges correspond to the non-zero entries. Nodes can be represented as variables (Li et al., 2023), diagonal entries of $A$ (Li et al., 2024) or omitted (Trifonov et al., 2024). A GNN then processes this graph to predict the non-zero entries of $L$, preserving the sparsity pattern of $A$.

Multiple rounds of message-passing aggregate information from neighboring nodes and edges. This architecture is local, meaning that if we modify a single entry of $A$, the change will propagate only to neighboring nodes and edges. The size of this neighborhood is limited by the GNN's receptive field, which is proportional to the network's depth, i.e., the number of message-passing layers. It is worth noting that this task is a regression on edges, which appear less frequently than typical node-level or graph-level tasks, for which most classical GNNs are formulated.

In principle, Cholesky factorizations admit formulations entirely in terms of message passing. Therefore they can be expressed as message-passing algorithms within modern GNN frameworks, analogous to the examples described in Moore et al. (2025). Such a message-passing realization would require network depth scaling with the problem size making the architecture impractically deep for realistic systems. Since our work focuses on learning approximate factors with GNNs without explicitly encoding the elimination procedure, we do not exploit this direct algorithmic realization.

**Related work**   The first attempts at learning sparse factorized preconditioners used convolutional neural networks (CNNs) (Sappl et al., 2019; Ackmann et al., 2020; Calì et al., 2023). However, CNNs are not feasible for large linear systems and can only be applied efficiently with sparse convolutions. Nevertheless,

sparse convolutions do not utilize the underlying graph structure of sparse matrices. Therefore, GNNs were naturally applied to better address the structure of sparse matrices, as has already been considered in several works (Li et al., 2023; Trifonov et al., 2024; Häusner et al., 2023; Li et al., 2024; Booth et al., 2024). GNNs can also be introduced into iterative solvers from other perspectives, not just for learning sparse incomplete factorizations. For example, the authors of Chen (2024) suggest using GNN as a nonlinear preconditioner, and in works Taghibakhshi et al. (2022; 2023) authors use GNNs for a domain decomposition method. Although, our work focuses on the sparse factorized preconditioners, the illustrated limitation is also relevant to the GNN-based approaches discussed above. Another line of work covers iterative solvers rather than preconditioners; see, for example, Luo et al. (2024).

The main limitations of GNNs, such as over-smoothing (Rusch et al., 2023), over-squashing (Topping et al., 2021) and expressive power bound (Xu et al., 2018) have already been highlighted in modern research. Mitigating these problems is indeed an important task, but in long-range dependent graphs, the receptive field must still be really large. Furthermore, deeper stacking of graph convolution layers may indeed work well for well-known GNNs benchmarks since they are typically sparse and connected, and one can expect a small graph diameter. However, for the problems considered in this paper, the diameter of the graphs can be up to $\mathcal{O}(N)$, making deeper stacking of layers infeasible.

### 2.1 Limitations of GNN-based Preconditioners

Consider the mapping $f : A \to L$, where $A$ is a given symmetric positive definite matrix, and $L$ is a sparse lower triangular matrix with a given sparsity pattern. In this section we will provide a an example of sparse matrices $A$, when:

- $A$ is a sparse matrix and there exists an ideal factorization $A = LL^\top$, where $L$ is a sparse matrix.

- The mapping of $A$ to $L$ is not local: a change in one entry of $A$ can significantly affect all entries of $L$, beyond the GNN's receptive field.

The simplest class of such matrices are positive definite tridiagonal matrices. These matrices appear from the standard discretization of one-dimensional PDEs. Such matrices are known to have bidiagonal Cholesky factorization

$$A = LL^\top, \tag{1}$$

where $L$ is bidiagonal matrix, and that is what we are looking for: the ideal sparse factorization of a sparse matrix. Our goal is to show that the mapping equation 1 is not local. Lets consider first the case of discretization of the Poisson equation on a unit interval with Dirichlet boundary conditions. The matrix $A$ is given by the second order finite difference approximation,

$$A = \begin{pmatrix} 2 & -1 & 0 & \cdots & 0 \\ -1 & 2 & -1 & \cdots & 0 \\ 0 & -1 & 2 & \ddots & \vdots \\ \vdots & \vdots & \ddots & \ddots & -1 \\ 0 & 0 & \cdots & -1 & 2 \end{pmatrix}. \tag{2}$$

The Cholesky factor $L$ is bidiagonal in this case.

The question is, how do the elements of $L$ change if we change a single entry of $A$ in position $(1, 1)$? The change in the diagonal is shown in Figure 1 on the left, where one can observe the decay. This decay is algebraic and aligned with the properties of the Green's functions of the PDEs. For this matrix, the dependence is local. However, we can construct pathological examples where the dependence is not local – a single change in $A$ will change almost all elements of $L$.

**Theorem 2.1.** *Let $A$ be a tridiagonal symmetric positive definite $n \times n$ matrix. Then it can be factorized as*

$$A = LL^\top,$$

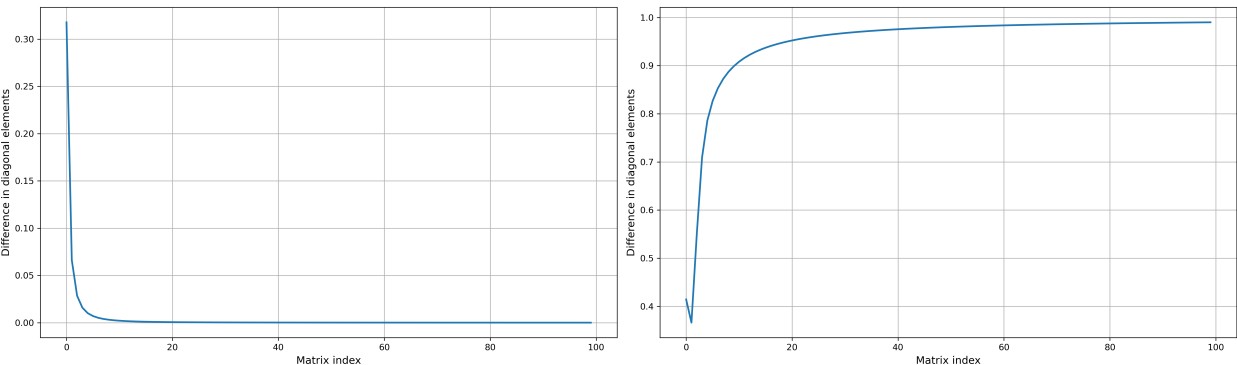

Figure 1: Difference of the diagonal elements between the Cholesky factor $L$ and perturbed factor $L'$ in a single entry $A_{11}$ of the tridiagonal matrix. **(Left)** 1D Laplacian matrix. **(Right)** Counterexample.

*where $L$ is a bidiagonal lower triangular matrix, and then mapping $A \to L$ is not local, which means that there exist matrix $A$ and $A'$ such that $A - A'$ has only one non-zero element, where the difference $L - L'$ has dense support: many entries change significantly, even though only a single entry was perturbed.*

*Proof.* Consider the matrix $A$ given by $A = LL^\top$ where $L$ is a bidiagonal matrix with $L_{ii} = \frac{1}{i}, i = 1, \ldots, n$ and $L_{i,i-1} = 1, i = 2$. Then $A$ is a symmetric positive definite tridiagonal matrix with elements $A_{11} = 1, A_{i,i} = 1 + \frac{1}{i^2}, A_{i+1,i} = A_{i,i+1} = \frac{1}{i}, i = 1, \ldots, n-1$. Now, consider the matrix $A' = A + e_1 e_1^\top$, where $e_1$ is the first column of the identity matrix. Let $A' = L'L'^\top$ be its Cholesky factorization. The matrix $L'$ is bidiagonal. The element $L'_{11}$ is equal to $\sqrt{2}$, and for each $i = 2, \ldots, n$ we have the well-known formulas

$$L'_{i,i-1} = \frac{L_{i,i-1}}{L'_{i-1,i-1}} = \frac{\frac{1}{i-1}}{L'_{i-1,i-1}},$$

and $L'_{i,i} = \sqrt{A_{i,i} - (L_{i,i-1})^2}$. Let $d_i = (L'_{i,i})^2$, then $d_1 = 2, d_i = 1 + \frac{1}{i^2} - \frac{1}{d_{i-1}(i-1)^2}$. From this recurrence relation it is easy to see that $d_i$ converges to 1 as $i \to \infty$. $\qquad\square$

Consequently, for a general matrix $A$, no finite-depth message passing architecture can robustly recover $L$ under such perturbations, so empirical failures are expected. The difference between diagonal elements of $L$ and $L'$ is shown on Figure 1 on the right.

## 3 Constructive Approach

We will use the class of tridiagonal matrices as the basis for our synthetic benchmarks for learning triangular preconditioners. What approaches can we take for other, more general sparse positive definite matrices? In this section, we present a constructive approach to building high-quality preconditioners that cannot be represented by GNNs, as demonstrated in our numerical experiments in Section 6.

To accomplish this task, we draw attention to the concept of K-condition number, introduced in Kaporin (1994). By minimizing this condition number, we can constructively build sparse preconditioners of the form $A \approx LL^\top$ for many matrices, where the sparsity pattern of $L$ matches that of the lower triangular part of $A$. The K-condition number of a symmetric positive definite matrix $A$ is defined as

$$K(A) = \frac{\frac{1}{n}\text{Tr}(A)}{(\det(A))^{1/n}}. \tag{3}$$

The interpretation of equation 3 is that it represents the arithmetic mean of the eigenvalues divided by their geometric mean. For matrices with positive eigenvalues, this ratio is always greater than 1 and equals 1 only

when the matrix is a multiple of the identity matrix. Given a preconditioned matrix $M$, we can evaluate its quality using $K(M)$. This metric can be used to construct *incomplete factorized inverse preconditioners* (Kaporin, 1994; Chen et al., 2024) $A^{-1} \approx LL^\top$ where $L$ is sparse. However, we focus on constructing *incomplete factorized preconditioners* $A \approx LL^\top$ with a sparse $L$. Therefore, we propose minimizing the functional:

$$K(L^\top A^{-1}L) \to \min_L, \tag{4}$$

where $L$ is a sparse lower triangular matrix with a predetermined sparsity pattern. Utilizing the inverse matrix in preconditioner optimization is a promising strategy that has been explored in other works (Li et al., 2023; Trifonov et al., 2024) through the functional:

$$\|LL^\top A^{-1} - I\|_F^2 \to \min. \tag{5}$$

The distinctive advantage of the functional equation 4 is that the minimization problem can be solved *explicitly* using linear algebra techniques. This allows us to construct pairs of matrices $(A_i, L_i)$ for small and medium-sized problems, where $L_i L_i^\top$ acts as an effective preconditioner. These pairs are valuable benchmarks for evaluating and comparing the properties of preconditioner learning algorithms against matrices that minimize equation 4.

## 4 K-optimal Preconditioner Based on Inverse Matrix for Sparse Matrices

In this section, we analyze the preconditioner quality functional:

$$K(L^\top A^{-1}L) \to \min_L, \tag{6}$$

where $L$ is a sparse lower triangular matrix with predetermined sparsity pattern. We will derive an explicit solution to this optimization problem.

### 4.1 Solution of the Optimization Problem

Let us demonstrate how to minimize the K-condition number in the general case, then apply the results to obtain explicit formulas for K-optimal preconditioners. Consider the optimization problem:

$$K(G^\top BG) \to \min_G, \tag{7}$$

where $G$ belongs to some linear subspace of triangular matrices:

$$g = \text{vec}(G) = \Psi z,$$

where $\Psi$ is an $n^2 \times m$ matrix, with $m$ being the subspace dimension. For sparse matrices, $m$ equals the number of non-zero elements in $G$.

Instead of directly minimizing functional equation 7, we minimize its logarithm:

$$\Phi(G) = \log K(G^\top BG) = \log \frac{1}{n}\text{Tr}(G^\top BG) - \frac{1}{n}\log \det(G)^2 - \frac{1}{n}\log \det(B).$$

The third term is independent of $G$ and can be omitted. For the first term:

$$\text{Tr}(G^\top BG) = \langle BG, G \rangle,$$

where $\langle \cdot, \cdot \rangle$ denotes the Frobenius inner product. Therefore:

$$\text{Tr}(G^\top BG) = (\mathcal{B}g, g),$$

with $\mathcal{B} = I \otimes B$, leading to:

$$\text{Tr}(G^\top BG) = (\mathcal{B}g, g) = (\Psi^\top \mathcal{B}\Psi z, z) = (Cz, z),$$

where $C = \Psi^\top \mathcal{B} \Psi$. To express the elements of matrix $C$, we use three indices for $\Psi$'s elements, $\Psi_{ii'l}$:

$$C_{ll'} = \sum_{i,j=1}^n B_{ij} \sum_{i'} \Psi_{ii'l} \Psi_{ji'l} = \langle B, \Psi_l \Psi_{l'}^\top \rangle,$$

where $\Psi_l, l = 1, \ldots, m$ are $n \times n$ matrices obtained from corresponding rows of $\Psi$. Our task reduces to minimizing with respect to $z$. Since $B$ is symmetric, $C$ is also symmetric, yielding the gradient:

$$\left(\nabla \Phi(z)\right)_j = \frac{2(Cz)_j}{(Cz, z)} - \frac{2}{n} \text{Tr}(G^{-1} \Psi_j),$$

derived using the formula for the logarithm of matrix determinant derivative.

**Special case: $G = L$ is a sparse matrix**  If $G = L$, where $L$ is a sparse lower triangular matrix, then matrix $C$ is a block-diagonal matrix of the form

$$C = \begin{pmatrix} C_1 & & & \\ & C_2 & & \\ & & \ddots & \\ & & & C_n \end{pmatrix},$$

where blocks $C_i$ are given by formulas

$$(C_i)_{kl} = B_{s_k^{(i)}, s_l^{(i)}},$$

where $s_k^{(i)}$ are indices of non-zero elements in the $i$-th column of matrix $L$, and matrix $G^{-1} \Psi_j$ has non-zero diagonal elements only for $j$ corresponding to diagonal elements of matrix $L$. For these elements $\text{Tr}(G^{-1} \Psi_j) = \frac{1}{g_{ii}}, i = 1, \ldots, n$.

The problem reduces to $n$ independent optimization problems on values of non-zero elements in the $i$-th column of matrix $L$, $i = 1, \ldots, n$. Let us consider each subproblem separately. The optimality condition for the $i$-th subproblem has the form

$$C_i z_i = \gamma_i e_1,$$

where $\gamma_i = \gamma_0 \frac{(Cz, z)}{g_{ii}}$ is a number, $\gamma_0$ is a constant that does not depend on $z$, $e_1$ is the first column of the identity matrix of corresponding size. Hence

$$z_i = \gamma_i v_i, \quad v_i = C_i^{-1} e_1,$$

and using the fact that $K$ does not depend on multiplication by a number we get an equation for the first component of vector $z$ (which is the diagonal element of matrix $L$)

$$(z_i)_1 = \frac{(v_i)_1}{(z_i)_1},$$

from which

$$(z_i)_1 = \sqrt{(v_i)_1}.$$

The vector $z_i$ contains the non-zero elements of $i$-th column of $L$. Therefore, the algorithm for finding the sparse lower triangular matrix $L$ is summarized in Algorithm 1. We refer to preconditioners constructed using this approach as *K-optimal preconditioners*.

---

**Algorithm 1** Construction of K-optimal preconditioner

---

**Require:** Symmetric positive definite matrix $A$, sparsity pattern for $L$
**Ensure:** Lower triangular matrix $L$
 1: Compute $B = A^{-1}$
 2: **for** $i = 1$ to $n$ **do**
 3:     Find indices $s_i$ of non-zero elements in column $i$ of $L$
 4:     Extract submatrix $B_i$ using rows and columns from $s_i$
 5:     Compute $v_i = B_i^{-1} e_1$                          ▷ $e_1$ is first unit vector
 6:     Set $L_{s_i,i} = ((v_i)_1)^{-1/2} \cdot v_i$          ▷ Store as $i$-th column of $L$
 7: **end for**

---

## 5 Benchmark Construction

To robustly evaluate the fundamental limitations of message-passing GNNs for sparse factorization, we constructed a two-part benchmark. One part is based on synthetic matrices engineered to have non-local dependencies. The other part is drawn from real-world sparse matrix problems. This dual approach ensures our benchmark covers both theoretically motivated and practically challenging instances.

### 5.1 Synthetic Benchmark

Our synthetic benchmark targets the core locality limitation of GNNs in a controlled setting. We generate tridiagonal matrices whose Cholesky factors $L$ are sparse but whose construction requires non-local information. Through empirical investigation, we found that simply fixing the diagonal elements of $L$ to 1 and sampling the off-diagonal elements from a normal distribution does not produce the desired non-local behavior, as the resulting matrices $A = LL^\top$ tend to exhibit primarily local dependencies. Our key insight is that non-locality emerges when the inverse matrix $L^{-1}$ is dense.

We exploit the property that the inverse of a bidiagonal matrix $L$ is rank-1 semiseparable, where the elements are given by the formula $L_{i,j}^{-1} = u_i v_j$ for $i \leq j$, representing part of a rank-1 matrix. This relationship is bidirectional, meaning that given vectors $u$ and $v$, we can construct $L^{-1}$ with this structure and then compute $L$ as its inverse. Our benchmark generation process exploits this property by randomly sampling appropriate vectors $u$ and $v$ to create matrices with guaranteed non-local dependencies. Poor performance on this benchmark would raise serious concerns about the fundamental suitability of current GNN architectures for matrix factorization tasks.

### 5.2 Matrices from the SuiteSparse Collection

To complement synthetic cases with practical relevance, we curated a collection of 150 symmetric positive definite matrices from the SuiteSparse matrix collection (Davis, 2024) for which dense inverse computation was feasible. For each, we computed K-optimal preconditioners by explicitly solving the optimization problem equation 7 under a fixed sparsity pattern matching the lower triangle of the original matrix, similar to incomplete Cholesky with zero fill-in (IC(0)) preconditioners (Saad, 2003). Our experimental results showed that K-optimal preconditioners generally outperformed traditional IC(0) preconditioners. In many cases, IC(0) either did not exist or required excessive iterations ($> 10000$) for convergence. However, we observed that for a small subset of matrices, IC(0) achieved better convergence rates.

The final benchmark consists of $(A_i, L_i)$ pairs, where each $A_i$ comes from SuiteSparse and $L_i$ represents factor of superior performing preconditioner, either IC(0) or K-optimal preconditioner. Note that constructing the K-optimal preconditioner requires access to the $A_i^{-1}$ so it is used here as an offline benchmark rather than as a practical method. Matrices where neither approach succeeded were excluded to ensure meaningful evaluation. The relative performance distribution between the K-optimal and IC(0) preconditioners is visualized in Figure 2. The K-optimal preconditioners are generally superior, though there are also cases in which IC(0) remains competitive or in which one or both methods fail to converge.

In our experiments, we consider only matrices for which a high-quality sparse preconditioner is known to exist—i.e., cases where the K-optimal method succeeds—so that the task remains feasible in principle and negative GNN results are meaningful.

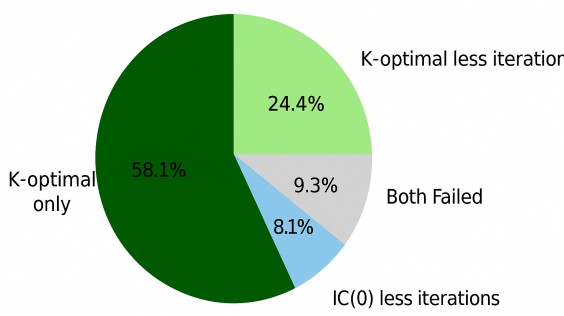

Figure 2: The performance of K-optimal preconditioner and IC(0) preconditioner during solution of SuiteSparse subset. **K-optimal only**: IC(0) preconditioner failed. **K-optimal less iterations**: both preconditioners were successful, with superior K-optimal performance. **IC(0) less iterations**: both preconditioners were successful, with superior IC(0) performance. **Both failed**: both preconditioners failed.

## 6 Experiments

### 6.1 GNN Architectures

Most classical GNNs either do not consider edges (e.g., GraphSAGE (Hamilton et al., 2017)) or consider them as scalar weighted adjacency matrix (e.g., Graph attention network (Veličković et al., 2017)). To enable edge updates during message-passing we use a Graph Network (Battaglia et al., 2018) block as a message-passing layer.

To validate GNNs on the proposed benchmarks, we use the popular Encoder-Processor-Decoder configuration. The Encoder consists of two separate MLPs, one for nodes and one for edges. The processor consists of multiple blocks of Graph Networks. First, the Graph Network updates the edge representations with Edge Model. Then, the nodes are update by the Node Model with a message-passing mechanism. In our work, we do not consider models that achieve a larger receptive field through graph coarsening or updates to graph-level information. Hence, the Global Model in the Graph Network is omitted. The decoder is a single MLP that decodes the hidden representations of the edges into a single value per edge in resulting factors $L$.

Graph coarsening strategies in GNN architectures can be naturally related to classical domain decomposition and multigrid methods. Moreover, these methods are well represented in the application of deep learning to numerical methods, e.g., works Taghibakhshi et al. (2022; 2023); Katrutsa et al. (2020); Luz et al. (2020). We intentionally omit coarsening techniques since their use would significantly shift the process of training incomplete factors toward other methods from classical preconditioner approaches (e.g., multigrid). Instead, to explicitly incorporate global information propagation within our setting, we employ the popular virtual node mechanism (Southern et al., 2024). This involves adding an extra node to the graph that is connected to every other node.

As a neural network baseline that does not propagate information between nodes, we use a simple two-layer MLP as the Node Model in Graph Network (MLPNodeModel). The following message-passing GNNs are used as the Node Model: (i) graph attention network v2 (GAT) (Brody et al., 2021), (ii) generalized aggregation network (GEN) (Li et al., 2020) and (iii) message-passing (MessagePassingMLP) (Gilmer et al., 2017) with two MLPs $f_{\theta_1}$ and $f_{\theta_2}$:

$$h_i = f_{\theta_2}\left(h_i, \frac{1}{N}\sum_{j \in \mathcal{N}(i)} f_{\theta_1}(h_i, e_{ij})\right),$$

Finally, we tested two graph transformers as Node Models: (i) graph transformer operator (GraphTransformer) from Shi et al. (2020) and (ii) global graph transformer operator (GlobalGraphTransformer) from Wu et al. (2023). The GlobalGraphTransformer integrates an all-pair node Attention Network (AN) with a Graph Network (GN) that preserves local graph structure:

$$\mathbf{Z}_{\text{out}} = (1 - \alpha)\,\text{AN}(\mathbf{Z}^{(0)}) + \alpha\,\text{GN}(\mathbf{Z}^{(0)}, \mathbf{A})\,,$$

where $\mathbf{Z}^{(0)}$ denotes the input node features and $\mathbf{A}$ the graph adjacency matrix. Following the original paper's design (Wu et al., 2023), we employ a simple graph convolution layer as the GN component (graph attention network v2 in ours experiments) and fix the mixing coefficient at $\alpha = 0.5$ in all experiments.

Note that the GlobalGraphTransformer achieves global all-pair attention with quadratic complexity in the number of nodes. This can be prohibitively expensive since medium-sized linear problems are of the size $N = 10^8$.

In our experiments, we set the encoders for nodes and edges to two layer MLPs with 16 hidden and output features. The Node Model is single-layer model from a following list: MLPNodeModel, GAT, GEN, MessagePassingMLP, GraphTransformer, GlobalGraphTransformer. The Edge Model is a two-layer MLP with 16 hidden features. The Node Model and Edge Model form the Graph Network, which is used to combine multiple message-passing layers in the Processor. The Edge decoder is a two-layer MLP with 16 hidden features and a single output feature. We use the all-ones vector $[1, \ldots, 1]^\top \in \mathbb{R}^N$ as the input node features, which was shown to be effective in Trifonov et al. (2024).

We trained the model for 300 epochs, reaching the convergence in each experiment. We used an initial learning rate of $10^{-3}$ and decreased it by a factor of 0.6 every 50 epochs. We also used early stopping with 50 epoch patience. For the synthetic dataset, we generated 1000 training samples and 200 test samples. The batch size was 16 and 8 for the synthetic and K-optimal datasets respectively.

We use the PyTorch Geometric (Fey & Lenssen, 2019) framework for training GNNs with main layers implementation. For the GlobalGraphTransformer, we use the source implementation from Wu et al. (2023). We used a single Nvidia A40 48Gb GPU for training.

To examine the importance of receptive field, we experiment with various numbers of layers within the Processor block. The maximum depth of the message-passing layers within the Processor block varies across different Node Models and is determined by GPU memory allocation for each Node Model but not greater than 7 layers.

## 6.2 Loss Function

For the GNN-based preconditioner, we can use training with objectives equation 4 and equation 5 to approximate true factor $L$ with $L(\theta)$ obtained by GNNs. Fortunately, we can avoid this since the optimization problem has an exact solution. Given the optimal factors $L$, the problem we want to solve is regression on the edges of the graph, for which the most natural loss is the $\mathcal{L}_2$ loss $\|L - L(\theta)\|_2$. However, we observed unstable training in our experiments and we could not achieve convergence with such a loss. Therefore, we switched to cosine similarity loss, which is well-suited to preconditioner learning. For sparse factorized preconditioners, the matrix $L$ is defined up to a scaling factor. For both preconditioners $X = LL^T$ and $\tilde{X} = (\alpha L)(\alpha L)^\top$ it is easy to show that the preconditioned systems $\tilde{X}^{-1}Ax = \tilde{X}^{-1}b$ will be equivalent since $\tilde{X}^{-1} = \frac{1}{\alpha^2}X^{-1}$ for $\alpha \neq 0$.

In general, a good preconditioner does not need to be an exact approximation of $A$, since iterative methods depend on the spectral properties of $X^{-1}A$. Besides empirical stability, the cosine similarity loss only penalizes mismatched direction but not mismatched scale. This allows for more possible variants of $L(\theta)$ and eases training. We expect a good approximation of $L$ in cosine similarity to be very close to 1.

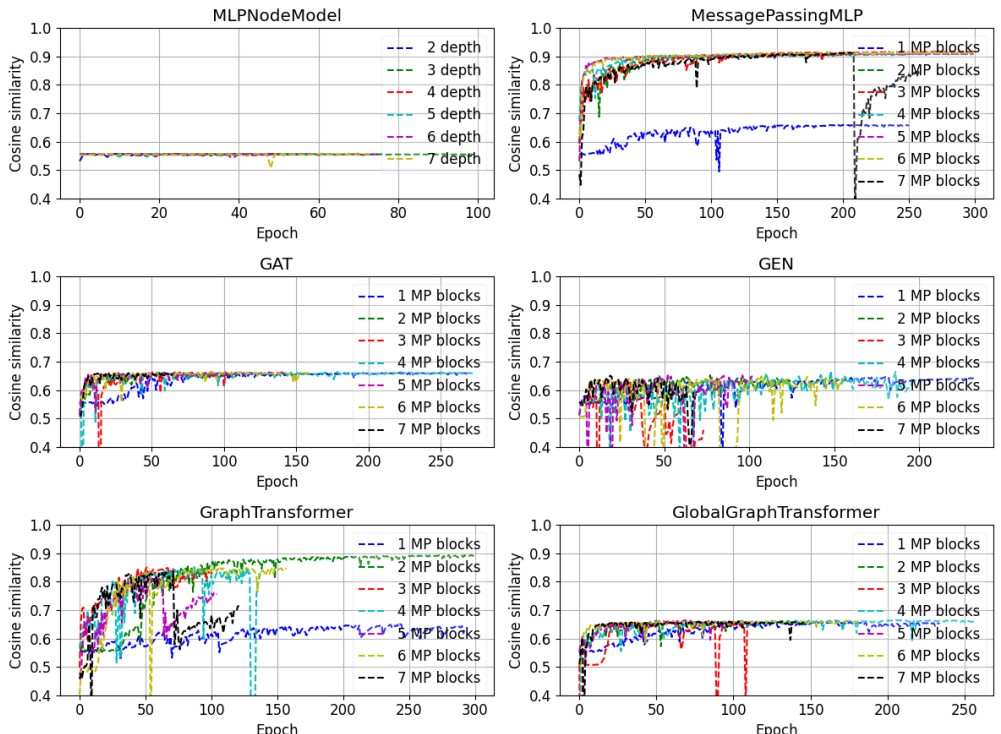

Figure 3: Experiments on the synthetic dataset. Cosine similarity between true $L$ and predicted $L(\theta)$ factors of preconditioners during training. Higher is better.

### 6.3 Learning Triangular Factorization

**Synthetic tridiagonal benchmark**  We begin our experiments with a synthetic benchmark, which is generated as described in Section 5.1. The modified training pairs $(A_i^m, L_i^m)$ are obtained as follows:

$$A_i^m = A_i + e_1 e_1^\top, \quad L_i^m = \text{chol}(A_i^m) . \tag{8}$$

where chol is a Cholesky factorization.

A trivial empirical justification of the non-local behaviour of the considered problem is performed using a deep feed-forward network, MLPNodeModel, which has no information about the neighbourhood context (Figure 3). Surprisingly, the classical graph network layers GAT and GEN achieve slightly higher final accuracy than MLPNodeModel. We assume that this behaviour is explained by the fact that these architectures are not designed to properly pass edge-level information, a primary goal of our work. The GNN with MessagePassingMLP layer, on the other hand, directly uses edge features, allowing it to produce satisfactory results with a number of rounds greater than one.

The GlobalGraphTransformer is designed for global all-pair attention. However, one can observe that straightforward global information propagation not necessarily lead to proper preconditioner approximation. Even global all-pair attention via node features does not enable the GlobalGraphTransformer to learn a good triangular factorization.

**K-optimal preconditioners on SuiteSparse subset**  Experiments with K-optimal preconditioners (Figure 4) show that none of the models can achieve a higher accuracy than the baseline feed-forward network, except MessagePassingMLP. However, MessagePassingMLP performs slightly better than the baseline model. Although GAT, GEN and GlobalGraphTransformer do not explicitly use edge features in their layers, infor-

mation should still propagate through the sender-receiver connection in the edge model. However, we did not observe any satisfactory approximation with these models.

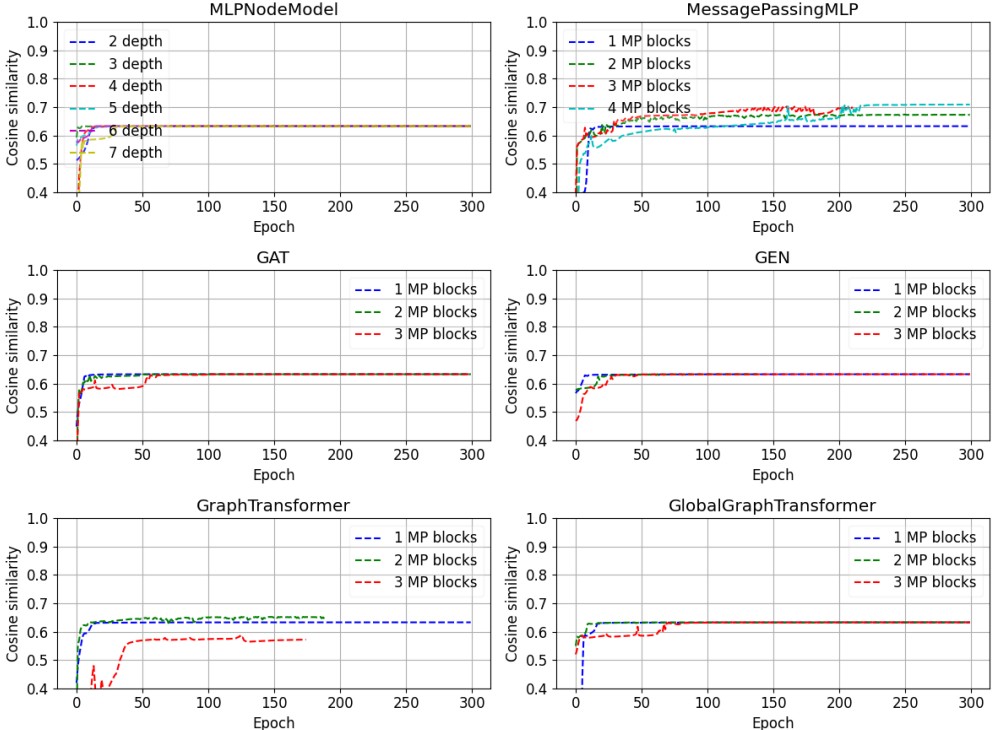

Figure 4: Experiments on the K-optimal preconditioners for the SuiteSparse subset. Cosine similarity between true $L$ and predicted $L(\theta)$ factors of preconditioners during training. Higher is better.

**Experiments with Virtual Node**  We also explored a simple yet popular form of explicit global information propagation by augmenting the graph with a virtual node, i.e., an additional node connected to every other node in the graph. This mechanism is often used in graph learning as a way to aggregate and redistribute global context. In our setting, however, it introduces a strong bottleneck: heterogeneous and multiscale features of the operator must pass through a single low-dimensional representation, which is closely related to the oversquashing phenomenon discussed in the GNN literature. Consistent with this intuition, our experiments show that adding a virtual node does not improve the quality of the learned preconditioners, while substantially increasing the number of edges and thus the computational cost. These results also suggest that naive global aggregation is insufficient to overcome the non-local dependencies inherent in high-quality triangular factors. The corresponding experimental results are reported in Appendix A.

**Order-dependence of the GNN-based preconditioners**  While a standard message-passing GNNs are permutation equivariant when applied to edge features of $A$ (Satorras et al., 2021), the overall mapping from $A$ to a triangular factor is inherently order-dependent. In our implementation, the GNN predicts values on the sparsity pattern of $A$, and we then construct the factor by retaining only $i \geq j$. It does not commute with arbitrary permutations. In general,

$$\mathrm{tril}(\mathrm{GNN}(P^\top A P)) \neq P^\top \mathrm{tril}(\mathrm{GNN}(A)) P\,.$$

Consequently, the learned preconditioner should be interpreted as order-dependent. This behavior is consistent with classical incomplete factorizations, which routinely employ reorderings to reduce fill to improve effectiveness. Rather than viewing the lack of strict permutation equivariance as a limitation, it motivates future work on incorporating ordering information more explicitly. The corresponding experimental results with positional encodings as in work Häusner et al. (2024) are reported in Appendix B.

# 7 Discussion

We have demonstrated that GNNs cannot recover the Cholesky factors for tridiagonal matrices, for which perfect sparse preconditioners exist. Under fully controlled conditions—where exact factorizations exist for synthetic matrices—the models failed to approximate the preconditioners. For the real-world matrices, one could argue that the GNNs can learn better preconditioners then K-optimal or IC(0) preconditioners in terms of quality. This is a subject of future work, but we believe that current benchmarks and the quality of computed preconditioners are quite challenging for state-of-the-art methods, even when using functionals other than cosine similarity. This claim is supported by the fact that current GNN-based preconditioners often do not outperform their classical analogues in terms of their effect on the spectrum of $A$ and are usually not universal.

For problems whose graphs can have very large diameter (up to $\mathcal{O}(N)$), an exact simulation of the underlying algorithm with message-passing would in principle require the same order of depth. In this regime, depths $r$ equals to 1 and 7 are both very small compared to what would be needed to show GNN the entire graph. There is no reason to expect performance to improve smoothly with depth. Instead, we observe an almost flat plateau corresponding to the best achievable local approximation, which persists until the depth becomes comparable to the graph diameter. In other words, we are likely approximating a function that does not admit increasingly accurate local approximations. Further work includes understanding the effective depth required by novel architectures, possibly with different aggregation mechanisms or global memory, to break the quality barrier we observe in our experiments.

A natural question is whether these limitations could be overcome by using deeper GNNs or architectures with global information propagation. However, our results show that even graph transformers with all-pairs attention and virtual node do not resolve the underlying non-locality barrier. This suggests that the problem is not only an insufficient receptive field, but also requires a dedicated architecture and shifts problem to designing architectures with the right algebraic inductive bias. The locality bias of message-passing is fundamentally incompatible with the non-local structure of sparse preconditioners.

These findings have two important implications. First, claims about GNNs' ability to learn matrix factorizations for scientific computing must be tempered by these structural limits. Second, progress will likely require architectural innovations that go beyond standard message-passing, potentially drawing insights from numerical linear algebra.

Another important design choice in classical sparse factorization is matrix reordering (Amestoy et al., 1996; Davis et al., 2004). Symmetric permutations can dramatically reduce fill-in and improve the structure of the elimination tree. In the present study, we keep the ordering fixed and focus on the difficulty of learning preconditioners for a given sparsity pattern. For our synthetic benchmark, the sparsity pattern is tridiagonal and the fill-in is already minimal by construction, so standard bandwidth-reducing permutations cannot further improve the structure. For real-world matrices, however, reorderings may alter the effective graph distances over which crucial dependencies propagate. Exploring the interaction between reordering strategies and learned preconditioners is an interesting direction for future work that could complement the limitations identified in this study.

While our experiments show that standard message-passing GNNs struggle to learn high-quality incomplete factorizations in the regimes we study, these findings should not be interpreted as a general failure of neural networks in scientific computing. The practical success of neural network-based solvers depends on many additional factors, including the structure of the matrix family under consideration, numerical conditioning, architectural and implementation choices, training strategies, etc. A deeper investigation of these aspects represents an important direction for future research in applying deep learning to numerical mathematics.

**Limitations** We restricted our attention to symmetric positive definite matrices. The section on the tridiagonal matrices remains unchanged, but the K-optimality does not apply to non-symmetric matrices. Hence, other approaches are necessary for constructing the corresponding benchmarks. We will address this issue in future work.

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

# A    Experimental Results with Virtual Node

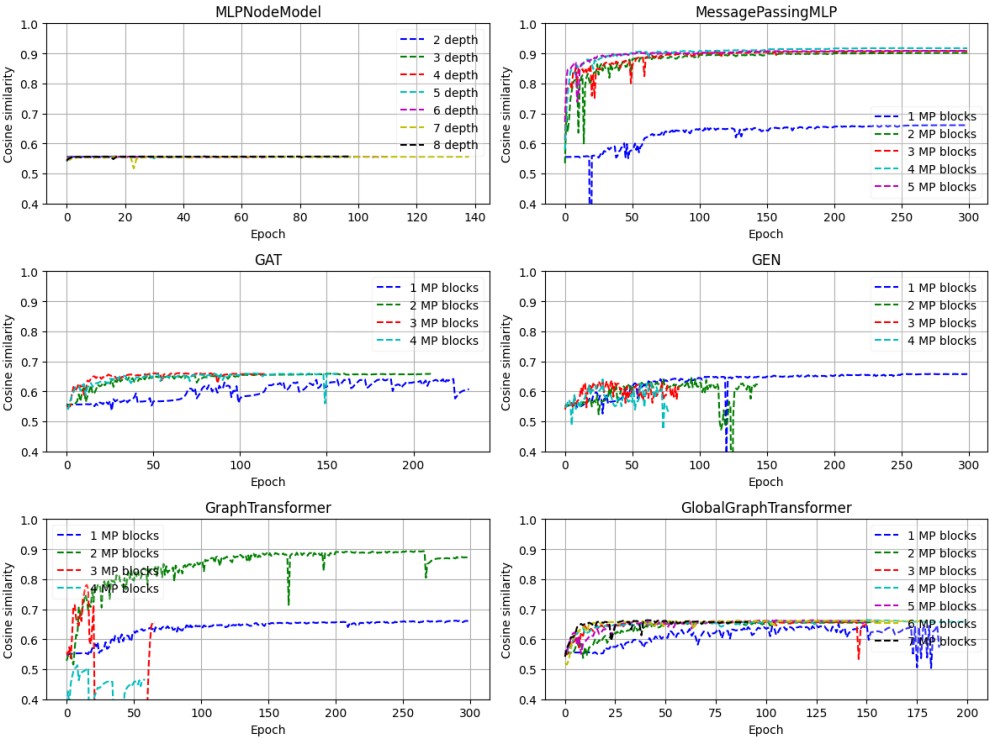

Figure 5: Experiments with virtual node on the synthetic dataset. Cosine similarity between true $L$ and predicted $L(\theta)$ factors of preconditioners during training. Higher is better.

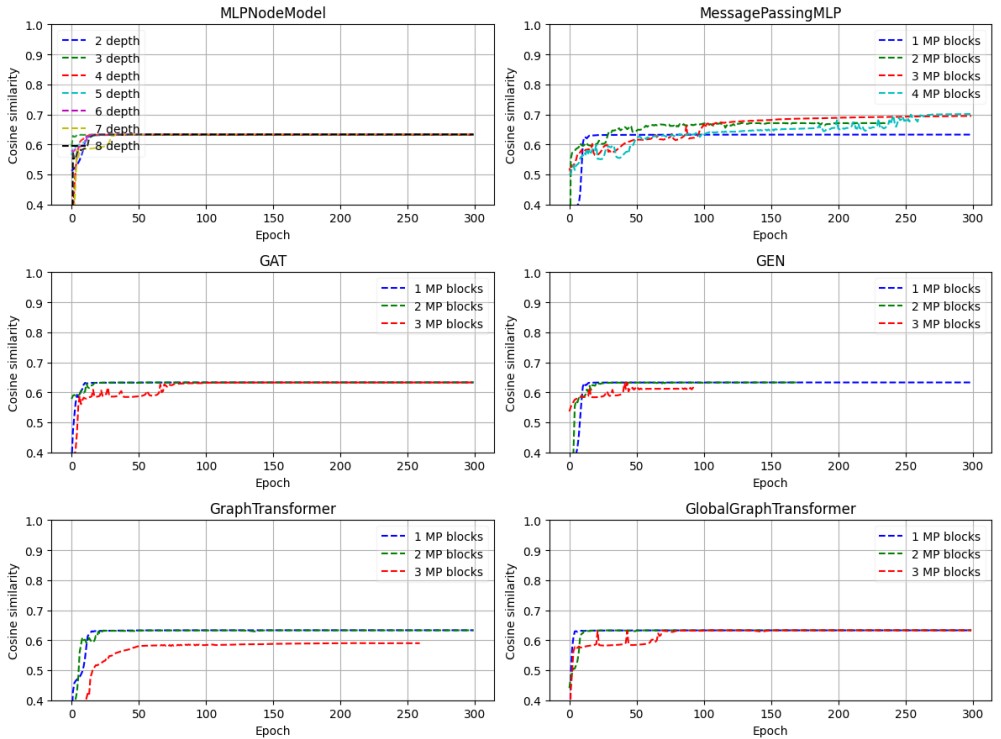

Figure 6: Experiments with virtual node on the K-optimal preconditioners for the SuiteSparse subset. Cosine similarity between true $L$ and predicted $L(\theta)$ factors of preconditioners during training. Higher is better.

## B Experimental Results with Positional Encodings

The positional encoding is an additional edge feature that has values $-1$ and $+1$ for the lower and upper triangular part of the matrix respectively (Häusner et al., 2024).

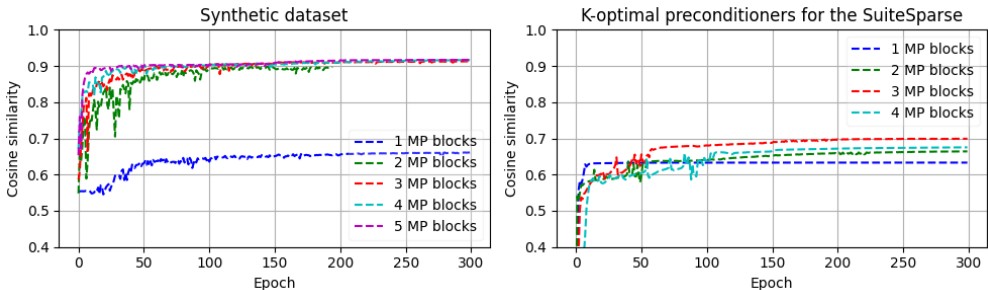

Figure 7: Experiments with positional encodings on the MessgaePassingMLP model. Cosine similarity between true $L$ and predicted $L(\theta)$ factors of preconditioners during training. Higher is better.

