# OpenReview forum: "Message-Passing GNNs Fail to Approximate Sparse Triangular Factorizations"
_TMLR — Accepted by TMLR_

### Review · Reviewer_Z8yc · 2025-11-12

**Summary Of Contributions:**

This paper investigates the fundamental limitations of message-passing GNNs in learning sparse triangular preconditioners for linear systems. The key contributions are: (i) theoretical and empirical evidence that non-local dependencies in high-quality preconditioners cannot be captured by standard message-passing architectures; (ii) a benchmark dataset (synthetic and SuiteSparse-derived) to evaluate GNN-based approaches; and (iii) systematic evaluation across multiple GNN variants showing consistently low approximation quality (cosine similarity ≤0.7).

**Strengths**: Clear motivation, technically sound methodology, rigorous mathematical analysis, and reproducible experimental design. The benchmark provides a valuable resource for the community.

**Areas for further elaboration**: The negligible performance gap between shallow and deep architectures needs some additional discussion. The potential role of matrix reordering techniques (e.g., AMD, COLAMD) in graph construction could also enhance the technical depth. (see Requested Changes)

**Audience:**

Yes

**Audience Explanation:**

The findings directly impact ML + scientific computing research by highlighting structural barriers in applying GNNs to matrix factorization. The benchmark enables future architectural innovation, making it relevant to both ML and numerical methods communities.

**Broader Impact Concerns:**

None. This work addresses a fundamental problem in numerical linear algebra with no apparent negative societal implications.

**Claims And Evidence:**

Yes

**Claims Explanation:**

The authors deliver compelling evidence through extensive evaluation across diverse matrix classes and GNN architectures. The theoretical framework correctly identifies non-local dependencies as a principal challenge. The benchmark construction follows standard practices and is carefully executed. The empirical findings could be further enriched by additional discussion of the observed depth-independence phenomenon, which would provide a more complete understanding of the architectural limitations.

**Requested Changes:**

**Important clarifications:**
The negligible performance gains from increased depth across most experiments (e.g., 1-layer vs. 7-layer models achieving nearly identical cosine similarity on SuiteSparse matrices) is surprising given the paper's emphasis on non-locality. The authors attribute this to "architectural mismatch," but this remains descriptive rather than diagnostic. The paper would be significantly strengthened by either (a) a hypothesis for why depth fails, supported by preliminary evidence (e.g., analyzing error patterns), or (b) explicitly framing this as an open problem for future work.

**Recommended enhancements:**
Consider discussing the potential role of matrix reordering techniques (e.g., AMD, COLAMD) in the graph representation. Since reordering is fundamental to classical sparse Cholesky factorization, examining whether directed graph structures or order-aware message passing could address some identified limitations would substantially enhance the paper's technical depth.

---

> ### Author Response · Authors · 2025-12-10
> **Authors' Answer to Reviewer Comments**
>
> We would like to thank the reviewer for the time to review our paper. Below we describe changes we incorporate into the revised manuscript to address reviewer's suggestions.
>
> > Important clarifications: The negligible performance gains from increased depth across most experiments (e.g., 1-layer vs. 7-layer models achieving nearly identical cosine similarity on SuiteSparse matrices) is surprising given the paper's emphasis on non-locality. The authors attribute this to "architectural mismatch," but this remains descriptive rather than diagnostic. The paper would be significantly strengthened by either (a) a hypothesis for why depth fails, supported by preliminary evidence (e.g., analyzing error patterns), or (b) explicitly framing this as an open problem for future work.
>
> We provide the following explanation in the revised manuscript.
>
> For problems whose graphs can have very large diameter (up to $\mathcal{O}(n)$), an exact simulation of the underlying algorithm with message-passing would in principle require the same order of depth. In this regime, depths $r$ equals to $1$ and $7$ are both very small compared to what would be needed to show GNN the entire graph. There is no reason to expect performance to improve smoothly with depth. Instead, we observe an almost flat plateau corresponding to the best achievable local approximation, which persists until the depth becomes comparable to the graph diameter. In other words, we are likely approximating a function that does not admit increasingly accurate local approximations. Further work includes understanding the effective depth required by novel architectures, possibly with different aggregation mechanisms or global memory, to break the quality barrier we observe in our experiments.
>
> > Recommended enhancements: Consider discussing the potential role of matrix reordering techniques (e.g., AMD, COLAMD) in the graph representation. Since reordering is fundamental to classical sparse Cholesky factorization, examining whether directed graph structures or order-aware message passing could address some identified limitations would substantially enhance the paper's technical depth.
>
> We thank the reviewer for valuable feedback. We provide the following explanation in the revised manuscript.
>
> Another important design choice in classical sparse factorization is matrix reordering [1-2]. Symmetric permutations can dramatically reduce fill-in and improve the structure of the elimination tree. In the present study, we keep the ordering fixed and focus on the difficulty of learning preconditioners for a given sparsity pattern. For our synthetic benchmark, the sparsity pattern is tridiagonal and the fill-in is already minimal by construction, so standard bandwidth-reducing permutations cannot further improve the structure. For real-world matrices, however, reorderings may alter the effective graph distances over which crucial dependencies propagate.
>
> Exploring the interaction between reordering strategies and learned preconditioners is an interesting direction for future work that could complement the limitations identified in this study.
>
> [1] Amestoy, Patrick R., Timothy A. Davis, and Iain S. Duff. "An approximate minimum degree ordering algorithm." SIAM Journal on Matrix Analysis and Applications 17.4 (1996): 886-905.
>
> [2] Davis, Timothy A., et al. "A column approximate minimum degree ordering algorithm." ACM Transactions on Mathematical Software (TOMS) 30.3 (2004): 353-376.

---

### Review · Reviewer_UMo9 · 2025-11-17

**Summary Of Contributions:**

This paper investigates a fundamental limitation of using graph neural networks (GNNs) to learn sparse matrix factorizations. The authors show that (approximate) factorizations of sparse matrices can exhibit non-local structure, making them challenging for local message-passing architectures to approximate. Using a constructed example based on tridiagonal matrices, they demonstrate that the exact bidiagonal Cholesky factorization can behave non-locally, which message-passing GNNs with a fixed number of steps are theoretically incapable of capturing. Empirically, they observe that GNNs do not recover the lower-triangular factors, as reflected by low cosine similarity between predictions and ground truth.

**Audience:**

Yes

**Audience Explanation:**

The work addresses a foundational question about the capabilities and limitations of GNN-based approaches for linear algebraic tasks, which should be of interest to the community.

**Broader Impact Concerns:**

There are no broader impact concerns for the paper.

**Claims And Evidence:**

No

**Claims Explanation:**

The theoretical results are interesting and relevant, but the empirical evaluation does not yet fully support the strong claims made in the paper. In particular, two standard architectural choices are not included in the analysis, and considering them could significantly strengthen the conclusions especially those involving the graph transformer baseline:
- **positional encodings**: GNNs and transformers are inherently permutation invariant so they do not know the output order which is, however, highly relevant for determining the lower triangular factor of the factorization. It would strengthen the paper to test whether positional encodings of each node, as in [1], mitigate some of the limitations.
- **virtual nodes**: according to the findings in the paper, the non-locality is an issue which could potentially easily resolved using a global node that is used to communicate between distant nodes which is a standard construction block in graph neural networks as recently studied for long-rage dependencies in [2], is it enough to add a single virtual node to learn the factorization?

Beyond these main concerns, I have several additional questions and suggestions that could help strengthen the empirical support for the claims.  I do not require the authors to address all of these points but clarifying some would be valuable:
- The theoretical results suggest that perfect performance requires O(n) steps. Given that, could you comment on why preconditioner performance does not improve when increasing the number of message-passing blocks beyond 2? One might expect incremental improvement even if exact recovery is not possible.
- Plotting performance as a function of the graph diameter would directly support the claim that locality is the limiting factor. Larger diameters should correlate with reduced performance for fixed depth. When depth equals the graph diameter, the paper’s claims are less applicable, so illustrating this boundary condition could add clarity.
- Could the authors elaborate why the experimental evaluation is conducted with respect to the K-optimal sparse matrices rather than the IC(0) matrices? Can GNNs learn IC(0) preconditioners? For the tridiagonal matrices, the IC(0) and K-optimal should coincide since the inverse is uniquely determined in the given sparsity pattern but for the suite sparse systems, computing the dataset is prohibitively expensive for large systems making the approach limited to small scale problems. What is the largest matrix in the suitsparse system that is considered?

[1] Häusner, Paul, Aleix Nieto Juscafresa, and Jens Sjölund. "Learning incomplete factorization preconditioners for GMRES", NLDL, 2025.
[2] Joshua Southern, Francesco Di Giovanni, Michael Bronstein, Johannes F. Lutzeyer. "Understanding virtual nodes: Oversmoothing, oversquashing, and node heterogeneity", ICLR, 2025.

**Requested Changes:**

Overall, the paper tries to answer an interesting and important question. The theoretical results indicate that previous methods have fundamental limitations that need to be addressed. However, the experimental section could more clearly support the full strength of the claims as I have outlined in my answer above. A number of clarifications and refinements would greatly improve readability and rigor:

- The condition number does not necessarily be small for preconditioning to work, rather the spectrum of the matrix is required to be clustered.
- The K-condition number needs a positive definite matrix so it is not clear how it is defined for $K(XA)$ in the text which is generally non-symmetric.
- Figure 2 may give the impression that computing the KKK-preconditioner is straightforward, although it requires access to $A^{−1}$; explaining this connection may help avoid confusion.
###### Questions
- How do eq. 5 and eq. 4 relate?
- Should Equation (8) involve the Cholesky factorization of the perturbed system rather than the original one?
- Does the K-optimal preconditioner always work when the IC(0) preconditioner works?
- Will the benchmark problems created for the paper be released?
- Because K-optimal factors are not unique, do you normalize them to make the learning target well-defined?
- What are the inputs to the node features of the GNN?
- What is the cosine similarity loss used for training? Does it only compare the L factors of the matrix M?
- What are the sizes of the considered matrices from the synthetic and suitesparse datasets?
###### Writing

The manuscript would benefit from improved clarity and consistent notation. Some suggestions:

- The notation for preconditioners shifts across sections: $X^{-1}$ in the introduction, $X$ in Section 3, lower-triangular $X$ in Section 4, and $M$ in Section 6. Unifying notation would improve readability.
- Referring to exact inverses of tridiagonal matrices as “preconditioners” may be confusing; using more precise terminology could help.
- Distinguish clearly between a _single_ message-passing step (local) and _multiple_ steps (expanded receptive field), particularly in the problem formulation.
- In Section 6.2, scaling the system does not produce an identical system, although it may not affect preconditioning performance; clarifying this point would be helpful.

Overall, I encourage the authors to provide additional technical details, ensure consistent notation, and refine the presentation for clarity.

---

> ### Author Response · Authors · 2025-12-10
> **Authors' Answer to Reviewer Comments**
>
> We would like to thank the reviewer for the time to review our paper. Below we describe changes we incorporate into the revised manuscript to address reviewer's suggestions. We thank the reviewer for additional comments about improved clarity and updated the revised manuscript accordingly.
>
> > GNNs and transformers are inherently permutation invariant so they do not know the output order which is, however, highly relevant for determining the lower triangular factor of the factorization. It would strengthen the paper to test whether positional encodings of each node, as in, mitigate some of the limitations.
>
> We thank the reviewer for the helpful suggestion and have updated the manuscript to reference the proposed work. We would like to clarify, however, that message-passing GNNs are permutation-equivariant for the problem of edge regression [1], which aligns well with the symmetry requirements of a mapping such as $A \rightarrow L(A) $:
>
> $$L(PAP^\top)=PL(A)P^\top$$
>
> The output triangular factor must transform consistently under permutations of the input matrix. Classical GNN architectures already enforce this equivariance by construction. By contrast, adding positional encodings tied to a specific node ordering would break permutation equivariance and make predictions dependent on the arbitrary input ordering. To preserve the desired symmetry, we do not employ order-dependent positional encodings in our experiments.
>
> > virtual nodes: according to the findings in the paper, the non-locality is an issue which could potentially easily resolved using a global node that is used to communicate between distant nodes which is a standard construction block in graph neural networks as recently studied for long-range dependencies in, is it enough to add a single virtual node to learn the factorization?
>
> Graph-level aggregation mechanisms can theoretically enable long-range communication. While this can be useful in some graph learning tasks, it introduces a strong information bottleneck: local heterogeneities and multiscale structures of the operator must be compressed into a single, fixed size global representation. In effect, we believe this can be considered as manually introducing an oversquashing bottleneck directly into the problem design, since potentially many independent local corrections are forced through a shared low-dimensional channel. Distinct local behaviors that should lead to different local preconditioning actions might be confused. For this reason, we do not view it as a robust or principled standalone strategy for constructing preconditioners with GNNs.
>
> To strengthen our claims, we added additional experiments with global information aggregation via a virtual node (node that is connected to each other node). In our experiments, the introduction of such a global interaction does not increase performance while still increasing the number of edges significantly.
>
> > why preconditioner performance does not improve when increasing the number of message-passing blocks beyond 2?
>
> We provide the following explanation in the revised manuscript.
>
> For problems whose graphs can have very large diameter (up to $\mathcal{O}(n)$), an exact simulation of the underlying algorithm with message-passing would in principle require the same order of depth. In this regime, depths $r$ equals to $1$ and $7$ are both very small compared to what would be needed to show GNN the entire graph. There is no reason to expect performance to improve smoothly with depth. Instead, we observe an almost flat plateau corresponding to the best achievable local approximation, which persists until the depth becomes comparable to the graph diameter. In other words, we are likely approximating a function that does not admit increasingly accurate local approximations. Further work includes understanding the effective depth required by novel architectures, possibly with different aggregation mechanisms or global memory, to break the quality barrier we observe in our experiments.
>
> > Plotting performance as a function of the graph diameter would directly support the claim that locality is the limiting factor.
>
> We agree that relating performance to graph diameter is conceptually appealing. However, we find it difficult to incorporate this suggestion. In our current benchmarks, graph diameter is strongly correlated with other properties (e.g., matrix size and structure), which makes it difficult to isolate its effect in a clean way. A careful, controlled study of how performance scales with diameter and other graph metrics would require additional analysis. We view this as an interesting direction for future work.
>
> [1] Satorras, Vıctor Garcia, Emiel Hoogeboom, and Max Welling. "E (n) equivariant graph neural networks." International conference on machine learning. PMLR, 2021.

---

> ### Author Response · Authors · 2025-12-10
> **Authors' Answer to Reviewer Comments, part 2**
>
> > Could the authors elaborate why the experimental evaluation is conducted with respect to the K-optimal sparse matrices rather than the IC(0) matrices? Can GNNs learn IC(0) preconditioners? For the tridiagonal matrices, the IC(0) and K-optimal should coincide since the inverse is uniquely determined in the given sparsity pattern but for the suite sparse systems, computing the dataset is prohibitively expensive for large systems making the approach limited to small scale problems.
>
> > Does the K-optimal preconditioner always work when the IC(0) preconditioner works?
>
> > Because K-optimal factors are not unique, do you normalize them to make the learning target well-defined?
>
> > What is the cosine similarity loss used for training? Does it only compare the L factors of the matrix M?
>
> As described in Section 5.2, we conduct experiments with a generally stronger preconditioner, namely the K-optimal preconditioner. According to Figure 2, the K-optimal preconditioner outperforms IC(0) in $82.5$\% of the cases, while IC(0) performs better in $8.1$\% of the cases (in the remaining $9.4$\% of the cases, both preconditioners fail). In $67.4$\% of the cases IC(0) preconditioner failed to construct.
>
> The K-optimal preconditioner is uniquely defined as the minimizer of the underlying optimization problem for a fixed sparsity pattern and lower-triangular structure. By contrast, IC(0) is uniquely defined purely by its construction and is not derived from an optimality criterion. Moreover, our learning objective is scale-invariant: we train using the negative cosine similarity between the vectorized target factors obtained from the factorizations and the corresponding factors predicted by the GNN. This loss eases optimization and ensures that any residual scaling ambiguity does not affect the training signal.
>
> > The condition number does not necessarily be small for preconditioning to work, rather the spectrum of the matrix is required to be clustered.
>
> This is indeed true. We added additional note on it in the revised version of the manuscript.
>
> > The K-condition number needs a positive definite matrix so it is not clear how it is defined for $K(XA)$ in the text which is generally non-symmetric.
>
> We thank the reviewer for this clarification. This is indeed a typo, we fixed it in the revised version of the manuscript.
>
> > Figure 2 may give the impression that computing the KKK-preconditioner is straightforward, although it requires access to $A^{-1}$; explaining this connection may help avoid confusion.
>
> We agree that this can introduce confusion. We added corresponding clarification.
>
> > How do eq. 5 and eq. 4 relate?
>
> Equations (4) and (5) are two different quality functionals defined on the same preconditioned operator. Equation (5) minimizes the Frobenius norm​, i.e., the least-squares distance to the identity, while equation (4) minimizes the K-condition number, which measures how tightly the eigenvalues of the preconditioned operator are clustered. We refer to the work [3] for greater details.
>
> > Should Equation (8) involve the Cholesky factorization of the perturbed system rather than the original one?
>
> It indeed should. Thank you for noticing this typo.
>
> > Will the benchmark problems created for the paper be released?
>
> Since the procedure for creating the synthetic benchmark and the algorithm for computing K-optimal preconditioners are straightforward, it is easy for others to reconstruct both datasets. Therefore, we do not plan to release the data separately.
>
> >What are the inputs to the node features of the GNN?
>
> Thank you for pointing out this omission, we updated the manuscript accordingly. We use the vector of all ones $[1, \dots, 1]^\top \in \mathbb{R}^n$ as proved efficient in work [2].
>
> > What are the sizes of the considered matrices from the synthetic and suitesparse datasets?
>
> The matrices in the synthetic benchmark have size $1000$. For the SuiteSparse collection, the minimum, maximum and median matrix sizes are $588$, $36 000$, $3 948$, respectively.
>
> [2] Trifonov, Vladislav, et al. "Learning from linear algebra: A graph neural network approach to preconditioner design for conjugate gradient solvers." arXiv preprint arXiv:2405.15557 (2024).
>
> [3] Kaporin, Igor E. "New convergence results and preconditioning strategies for the conjugate gradient method." Numerical linear algebra with applications 1.2 (1994): 179-210.

---

> > ### Comment · Reviewer_UMo9 · 2025-12-10
> >
> > I would like to thank the authors for their response to my questions. However, my concern regarding the permutation equivariance is not fully resolved yet. I will try to outline my understanding of the method and point to the arising issue within this framework. Please correct me if there is some missunderstanding how the lower triangular factor is obtained.
> >
> > As I understand, the authors first apply a message-passing neural network f to the input matrix A and then use the lower triangular part of this output as the triangular factor i.e. $L = tril(f(A))$ however, the exact way how this is implemented in the method is not described in the text of the manuscript. While the message-passing steps are permutation equivariant as pointed out by the reviewers, this does not hold for taking the lower triangular part of the matrix. This can be easily seen since:
> >
> > $tril(f(P^TAP)) \neq P^Ttril(f(A))P$
> >
> > the matrix on the left-hand side is lower triangular by construction. However, after permuting the rows and columns on the right-hand side the matrix is not lower triangular any more and thus the two sides can not be equivalent and the permutation equivariance is broken.

---

> > > ### Author Response · Authors · 2025-12-12
> > > **Official Comment by Authors**
> > >
> > > We appreciate the reviewer’s insistence on this point, which helped us identify and correct an inaccurate claim in an earlier comment regarding the relation $L(P^\top A P) \neq P^\top L(A) P$. This clarification removed a source of confusion. It also highlights an important open direction: whether a learned preconditioning operator should in principle exhibit permutation equivariance, given that classical incomplete factorizations deliberately exploit reordering to improve their quality.
> > >
> > > The positional encodings suggested by the reviewer are particularly relevant for methods that explicitly predict both $L$ and $U$ factors, as in [1]. In our approach, the GNN outputs values on the sparsity pattern of $A$ and in the final step we retain only edge entries with $i \ge j$. Nevertheless, exploring positional encodings tailored to learning sparse incomplete factorization preconditioners remains a compelling direction for future work.
> > >
> > > [1] Häusner, Paul, Aleix Nieto Juscafresa, and Jens Sjölund. "Learning incomplete factorization preconditioners for GMRES", NLDL, 2025.

---

### Review · Reviewer_ri4G · 2025-11-26

**Summary Of Contributions:**

The paper’s title, “Message-Passing GNNs Fail to Approximate Sparse Triangular Factorizations”, summarizes the paper’s main content well. The paper studies the recent trend of approximating the inverse of a sparse matrix with a graph neural network. Noting that the inverse of a sparse matrix may introduce global dependencies among its nonzero entries, the paper constructs synthetic and practical matrices to demonstrate that existing GNN frameworks struggle to learn a good approximation of matrix factorization. Based on the experiment results, the paper concludes that existing GNNs have major structural limitations for learning matrix factorization in scientific computing and suggests that future research in this area requires innovative designs in the network architecture.

**Audience:**

Yes

**Audience Explanation:**

Yes, I believe the topic of this paper will be of interest to researchers working on AI for linear solvers/PDEs. While multiple papers have attempted to develop network methods for solving matrices, few have conducted comprehensive comparisons between neural network solvers and classical solvers on large-scale problems. In fact, many classical solvers remain highly competitive, or even superior, in solving large-scale problems [1]. This TMLR submission makes a good effort to calibrate our expectations for a network method in solving classic numerical problems. Although it does not directly address solving PDEs/preconditioning linear systems (approximating the inverse of a matrix is, of course, highly relevant), the research direction it advocates for is critical for the healthy development of AI for linear solvers/PDEs.

[1] Weak baselines and reporting biases lead to overoptimism in machine learning for fluid-related partial differential equations. Nature Machine Intelligence 2024.

**Broader Impact Concerns:**

None.

**Claims And Evidence:**

Yes

**Claims Explanation:**

The paper’s motivation is well grounded in known facts in applied math, e.g., the properties of the factorization of a tridiagonal matrix, the Green's function of Poisson’s equation, and the closed-form solution to Kaporin preconditioning. The central claim is well motivated and follows naturally from these known facts. Overall, based on my research experience in this area over the past few years, I am generally OK with most of the claims and the experimental evidence reported in the paper.

A few claims in the paper could benefit from further clarification, which I list in the requested changes below.

**Requested Changes:**

Overall, this is good work. I have only two things to suggest:

1. The derivation of the closed-form solution for the Kaporin preconditioner in Section 4 reminds me of results reported in previous work, particularly Appendix A of Chen et al. [2]. Please clarify the novelty of the results in Section 4 relative to it.

2. While the paper provides convincing evidence that GNNs struggle to approximate matrix factorizations in some instances, it occasionally generalizes such limitations to broader scientific computing problems. In practice, many other factors play critical roles in determining whether a network can actually work well for a problem, e.g., the matrix pattern, implementation details, and system engineering/optimization, which are not fully covered in this paper’s discussion. I suggest the paper also inform readers of these factors and refrain from overgeneralizing their findings to (general) scientific computing before we see a more comprehensive analysis, which is perhaps outside the scope of this work.

[2] “Lightning-fast method of fundamental solutions” (ACM TOG 2024) by Chen et al.

---

> ### Author Response · Authors · 2025-12-10
> **Authors' Answer to Reviewer Comments**
>
> We would like to thank the reviewer for the time to review our paper. Below we describe changes we incorporate into the revised manuscript to address reviewer's suggestions.
>
> > The derivation of the closed-form solution for the Kaporin preconditioner in Section 4 reminds me of results reported in previous work, particularly Appendix A of Chen et al. Please clarify the novelty of the results in Section 4 relative to it.
>
> The derivation presented in work [1], Appendix A traces back to the earlier results of [2], Appendix A.3. The main difference of the results in Section 4 of our manuscript is that [1, 2] derive results for *incomplete inverse Cholesky preconditioning*, i.e., factors $L$ satisfying $LL^\top \approx A^{-1}$. In contrast, prior works on learning incomplete factorizations with GNNs and our paper focus on constructing preconditioners of the *incomplete Cholesky (IC) type*, specifically IC with zero fill-in, where the factors $L$ satisfy $LL^\top \approx A$.
>
> We thank the reviewer for the helpful suggestion and have updated the manuscript to reference the proposed work.
>
> > I suggest the paper also inform readers of these factors and refrain from overgeneralizing their findings to (general) scientific computing before we see a more comprehensive analysis, which is perhaps outside the scope of this work.
>
> We thank the reviewer for this suggestion. We have added a corresponding paragraph to the Discussion section that explicitly acknowledges these broader factors and clarifies that our claims apply only to the specific learning regime studied, thereby softening any unintended generalization of our findings.
>
> [1] Chen, Jiong, Florian T. Schäfer, and Mathieu Desbrun. "Lightning-fast method of fundamental solutions." ACM Transactions on Graphics 43.4 (2024): 77.
>
> [2] Kaporin, Igor E. "New convergence results and preconditioning strategies for the conjugate gradient method." Numerical linear algebra with applications 1.2 (1994): 179-210.

---

### Review · Reviewer_kLEU · 2025-11-26

**Summary Of Contributions:**

There are two main contributions of this work. First, a dataset of SPD matrices for which an “optimal” (in the K-condition number sense) preconditioner can be found with the same sparsity pattern as the original matrix. This dataset contains two types of matrices: a synthetic dataset of tridiaganal matrices and a set of matrices from the SuiteSparse collection. In both cases, the matrices are such that an “optimal” preconditioner can be found, thereby giving the graph neural networks a target to learn.


The second contribution is experimental evidence that graph neural networks (GNNs) cannot be used to approximate such preconditioners. Several common graph neural network layers are trained in encoder-process-decode models to maximize cosine similarity. In general, the experiments provide evidence that in cases where good preconditioner construction requires non-local information, GNNs are not able to approximate such optimal preconditioners.

**Additional Comments:**

- From a grammatical perspective, all important words in titles of sections, subsections, etc. should be capitalized.
- Typo Page 1: “A well-established choice for symmetric positive definite matrices is to use an incomplete Cholesky **factorization**
- Typo Page 3: line after eq (1) “biadiagonal” should be **bidiagonal**

**Audience:**

Yes

**Audience Explanation:**

Construction of effective preconditioners for linear systems is an ongoing research area with broad appeal and the potential to improve computational science along with many other scientific areas which depend on efficient solutions to large, sparse linear systems. The construction of an effective preconditioner is highly problem-dependent. Therefore, the hope is that machine learning methods can provide a method which is able to construct preconditioners from the entries of the matrix. As the author(s) mention, this has been attempted in prior literature and while the preconditioners show some promise, most fall short of well -tuned, human-designed preconditioners. One of the strengths of this paper is the construction of a database of problems for which good preconditioners are known, but that current GNN methods are unable to replicate (at least in practical use, see my note in the “requested changes” response.) This should give researchers in this area a benchmark that future methods and architectures can target as a proxy for problems where non-local information is required for building effective preconditioners.

**Broader Impact Concerns:**

None.

**Claims And Evidence:**

Yes

**Claims Explanation:**

In general, the analysis for the fact that a change in a single entry of a matrix can significantly affect many entries of the corresponding Cholesky factor appears accurate, as does the construction approach for building K-optimal preconditioners. The construction of the synthetic dataset appears sound, both from a theoretical standpoint and supported by the computational experiments. Based on the completed experiments, both components of the suggested dataset do appear to be adequate problems for testing how well a potential model may be able to approximate a preconditioner of the type studied in this work (an approximate Cholesky factorization with the same sparsity pattern as the original matrix). That being said, I do have a couple of minor concerns which is given in the “requested changes” response below.

**Requested Changes:**

While I agree that the computational experiments presented in the paper provide evidence from a practical standpoint that GNNs can’t approximate K-optimal preconditioners, I disagree that this proves that GNNs can’t, at least in theory, compute these factorizations. Since the calculation of a Cholesky factorization can be completed using matrix multiplications, I believe the methods described in [1] could be used to construct a GNN which computes a Cholesky factorization exactly. While this wouldn’t be sparse for a general SPD matrix, it would of course be sparse for the tridiagonal matrices in the synthetic dataset.

This counters the author’s assertion that widening the receptive field is not enough to be able to construct an effective preconditioner. However, the resulting GNN would have more layers than the number of rows in the matrix. Therefore, the required theoretical GNN would be unreasonably large and even if it was feasible to use such a network, the probability that an optimizer could locate the precise weights needed to attain the necessary theoretical network would be small.

Therefore I don’t believe the experiments in the paper are misleading or incorrect and I agree with the author(s) that new architectures are needed if reasonably-sized GNNs are to be used to construct effective preconditioners, but the discrepancy between the theoretical possibility and the practical implementation needs to be addressed.

Also, graph algorithms that enlarge the receptive field by coarsening or updating graph-level information aren’t considered at all, but no explanation is given for why they are not considered. If the reason they are disregarded is out of belief that the size of the receptive field is immaterial, that argument is more cloudy given the argument above. Thus some experimental evidence would be useful for firming up the argument that these methods are still insufficient.

[1] Moore NS, Cyr EC, Ohm P, Siefert CM, Tuminaro RS, "Graph Neural Networks and Applied Linear Algebra", SIAM Rev, 2025

---

> ### Author Response · Authors · 2025-12-10
> **Authors' Answer to Reviewer Comments**
>
> We would like to thank the reviewer for the time to review our paper. Below we describe changes we incorporate into the revised manuscript to address reviewer's suggestions. We also thank the reviewer for indicating grammatical issues.
>
> > Note regarding theoretical possibility and the practical implementation about expression of Cholesky factorization as GNN.
>
> We agree with the reviewer that both exact Cholesky factorizations and algorithms with predefined sparsity patterns (i.e., IC(0) and K-optimal preconditioners) admit formulations entirely in terms of message passing. Therefore they can be expressed as message-passing algorithms within modern GNN frameworks, including the examples described in [1]. Since our work focuses on learning approximate factors with GNNs without explicitly encoding the elimination procedure, we do not exploit this direct algorithmic realization.
>
> We also agree that the distinction between what trained GNNs can be expected to learn in practice and what they can express in theory should be made explicit. We have added this clarification to the revised manuscript.
>
> > Why the receptive field by coarsening or updating graph-level information isn't considered at all.
>
> We thank the reviewer for indicating an omission of this discussion. Moreover, we conducted additional experiments with global information propagation by means of virtual node. We introduced these changes in the revised manuscript. Let us also clarify our reasoning in this comment:
>
> Graph coarsening strategies in GNN architectures can be naturally related to classical domain decomposition and multigrid methods. Moreover, these methods are well represented in the application of deep learning to numerical methods, e.g.,  works [2-5]. We intentionally omit coarsening techniques since their use would significantly shift the process of training incomplete factors toward other methods from classical preconditioner approaches (e.g., multigrid).
>
> Graph-level aggregation mechanisms can theoretically enable long-range communication. While this can be useful in some graph learning tasks, it introduces a strong information bottleneck: local heterogeneities and multiscale structures of the operator must be compressed into a single, fixed size global representation. In effect, we believe this can be considered as manually introducing an oversquashing bottleneck directly into the problem design, since potentially many independent local corrections are forced through a shared low-dimensional channel. Distinct local behaviors that should lead to different local preconditioning actions might be confused. For this reason, we do not view it as a robust or principled standalone strategy for constructing preconditioners with GNNs.
>
> To strengthen our claims, we added additional experiments with global information aggregation via a virtual node (node that is connected to each other node) [6]. In our experiments, the introduction of such a global interaction does not increase performance while still increasing the number of edges significantly.
>
> While in this work we focus solely on training incomplete factors, we believe that both of the approaches described above have to be systematically studied in order to achieve high quality neural network-constructed preconditioners.
>
> [1] Moore NS, Cyr EC, Ohm P, Siefert CM, Tuminaro RS, Graph Neural Networks and Applied Linear Algebra, SIAM Review, 2025.
>
> [2] Taghibakhshi, Ali, et al. "Mg-gnn: Multigrid graph neural networks for learning multilevel domain decomposition methods." International conference on machine learning. PMLR, 2023.
>
> [3] Taghibakhshi, Ali, et al. "Learning interface conditions in domain decomposition solvers." Advances in Neural Information Processing Systems 35 (2022): 7222-7235.
>
> [4] Katrutsa, Alexandr, Talgat Daulbaev, and Ivan Oseledets. "Black-box learning of multigrid parameters." Journal of Computational and Applied Mathematics 368 (2020): 112524.
>
> [5] Luz, Ilay, et al. "Learning algebraic multigrid using graph neural networks." International Conference on Machine Learning. PMLR, 2020.
>
> [6] Joshua Southern, Francesco Di Giovanni, Michael Bronstein, Johannes F. Lutzeyer. "Understanding virtual nodes: Oversmoothing, oversquashing, and node heterogeneity", ICLR, 2025.

---

### Author Response · Authors · 2026-02-24
**Acknowledgment of Reviewers and Action Editors Feedback**

Dear Reviewers and Action Editors,

We thank you again for your thorough work in reviewing our paper. We have included an additional paragraph, as well as experiments with positional encodings, in the final version of the manuscript.

Best regards,
Authors

---

### Decision · Action_Editor_C2AG · 2026-01-31

**Recommendation:** Accept with minor revision

**Additional Comments:**

Given the active discussion on the permutation equivariance of learned preconditioning operators, I recommend to add further discussion as well as a small ablation study on using positional encodings (e.g., for the MessagePassingMLP variant) in the final version of the paper.

**Audience:**

Yes

**Audience Explanation:**

The results are relevant to the scientific ML community.

**Claims And Evidence:**

Yes

**Claims Explanation:**

The paper provides convincing evidence, combining rigorous mathematical reasoning and numerical evaluation with a new standardized benchmark.